# Mass Spectrometry-Based Proteomic Discovery of Prognostic Biomarkers in Adrenal Cortical Carcinoma

**DOI:** 10.3390/cancers13153890

**Published:** 2021-08-02

**Authors:** Han Na Jang, Sun Joon Moon, Kyeong Cheon Jung, Sang Wan Kim, Hyeyoon Kim, Dohyun Han, Jung Hee Kim

**Affiliations:** 1Department of Internal Medicine, Seoul National University College of Medicine, Seoul 03080, Korea; chhanna@snu.ac.kr (H.N.J.); sj.md.moon@samsung.com (S.J.M.); swkimmd@snu.ac.kr (S.W.K.); 2Department of Internal Medicine, Seoul National University Hospital, Seoul 03080, Korea; 3Division of Endocrinology and Metabolism, Department of Internal Medicine, Kangbuk Samsung Hospital, Sungkyunkwan University School of Medicine, Seoul 03080, Korea; 4Department of Pathology, Seoul National University College of Medicine, Seoul 03080, Korea; jungkc66@snu.ac.kr; 5Department of Translational Medicine, Seoul National University College of Medicine, Seoul 03080, Korea; 6Integrated Major in Innovative Medical Science, Seoul National University Graduate School, Seoul 03080, Korea; 7Department of Internal Medicine, Seoul Metropolitan Government Seoul National University Boramae Medical Center, Seoul 03080, Korea; 8Proteomics Core Facility, Biomedical Research Institute Seoul National University Hospital, Seoul 03080, Korea; hyeyoonk@snu.ac.kr

**Keywords:** adrenal cortical carcinoma, prognosis, proteomics, mass spectrometry, HNRNPA1

## Abstract

**Simple Summary:**

Adrenal cortical carcinoma (ACC) is an extremely rare disease with a variable prognosis. Current prognostic markers have limitations in identifying patients with a poor prognosis and who require adjuvant therapy. We developed the prognostic biomarker candidates of ACC using mass-spectrometry-based proteomics and machine learning algorithm. We further validated them in The Cancer Genome Atlas data and performed the survival analysis according to the expression levels of each protein. In addition, HNRNPA1, the protein identified as a prognostic marker of ACC based on proteomics, was validated in the immunohistochemistry staining. The prognostic protein biomarkers of ACC found in this study are expected to help determine the appropriate treatment plan for patients.

**Abstract:**

Adrenal cortical carcinoma (ACC) is an extremely rare disease with a variable prognosis. Current prognostic markers have limitations in identifying patients with a poor prognosis. Herein, we aimed to investigate the prognostic protein biomarkers of ACC using mass-spectrometry-based proteomics. We performed the liquid chromatography–tandem mass spectrometry (LC–MS/MS) using formalin-fixed paraffin-embedded (FFPE) tissues of 45 adrenal tumors. Then, we selected 117 differentially expressed proteins (DEPs) among tumors with different stages using the machine learning algorithm. Next, we conducted a survival analysis to assess whether the levels of DEPs were related to survival. Among 117 DEPs, HNRNPA1, C8A, CHMP6, LTBP4, SPR, NCEH1, MRPS23, POLDIP2, and WBSCR16 were significantly correlated with the survival of ACC. In age- and stage-adjusted Cox proportional hazard regression models, only HNRNPA1, LTBP4, MRPS23, POLDIP2, and WBSCR16 expression remained significant. These five proteins were also validated in TCGA data as the prognostic biomarkers. In this study, we found that HNRNPA1, LTBP4, MRPS23, POLDIP2, and WBSCR16 were protein biomarkers for predicting the prognosis of ACC.

## 1. Introduction

Adrenal cortical carcinomas (ACCs) are rare tumors with an annual incidence of 0.7–2.0 cases per million [1,2]. The overall prognosis of ACC is poor, with a 5-year survival rate of less than 40% [2,3]. However, the prognosis of ACCs varies: resectable tumors have a good prognosis, while metastatic and unresectable tumors have a poor prognosis. The European Network for the Study of Adrenal Tumors (ENSAT) tumor staging and tumor grading, such as Ki-67 proliferation index or mitotic count, have been suggested as prognostic factors [4,5,6,7]. However, up to 70% of patients with the localized disease based on TNM staging experienced recurrence within 3 years after surgery [3], while the survival duration of patients with metastatic ACCs ranges from several months to 10 years or more [8]. Therefore, these prognostic factors still have limitations in predicting the prognosis of ACCs.

Several efforts were made to assess the molecular prognostic markers for ACC [9,10,11,12]. Comprehensive pan-genomic characterization of ACCs using the ENSAT network and The Cancer Genome Atlas (TCGA) led to the identification of new distinct molecular subtypes [13,14]. However, the use of pan-genomic measures in clinical practice is limited as they are complex and expensive. In addition, a recent meta-analysis of pan-genomic studies showed that molecular classification could be helpful in distinguishing localized ACCs, but its performance in metastatic ACCs remained questionable [15].

The proteomic approach has rarely been used to study ACCs. In The Cancer Proteome Atlas (TCPA) database, the protein expression of 46 ACCs was measured using the reverse-phase protein array (RPPA), which is an antibody-based quantitative method for assessing multiple protein markers [16]. Our group obtained proteomics data from the TCPA database and demonstrated that overexpression of cyclin B1, transferrin receptor 1, and fibronectin is associated with poor prognosis of ACCs [12]. However, the RPPA method identified only 198 proteins.

Liquid chromatography–tandem mass spectrometry (LC–MS/MS) is a cutting-edge technology that analyzes several thousand proteins rapidly and accurately with high sensitivity [17]. To the best of our knowledge, only one study has performed an LC–MS/MS analysis to distinguish the differentially expressed proteins (DEPs) of eight ACCs from six adrenal cortical adenomas (ACAs) [9]. However, this study did not elucidate the prognostic factors of ACCs.

Fresh frozen samples are often preferred for molecular profiling, as macromolecules are preserved without cross-links [18], but the availability of fresh frozen samples is often limited because their collection is laborious, is expensive, and requires special logistics. Hence, most human tissue specimens archived in hospitals for routine diagnostic purposes are formalin-fixed paraffin-embedded (FFPE) blocks, which are stored for a long time at room temperature without quality reduction and are usually associated with rich clinical and phenotypic data, including histology, diagnosis, treatment history, response, and outcome [19]. Recent technical developments in MS-based proteomics methods and protein extraction protocols have enabled the in-depth proteomic analysis of FFPE tumor tissues [20]. Previous studies reported that FFPE tissues show qualitative and quantitative proteomic properties similar to those of fresh frozen tissues, further highlighting the potential of FFPE tissue analysis for biomedical research [21].

Here, we aimed to use LC–MS/MS to analyze FFPE tumor tissues and to elucidate the protein markers for predicting the prognosis of ACCs.

## 2. Materials and Methods

### 2.1. Study Design

The Seoul National University Hospital (SNUH) cohort was comprised of 37 patients with ACC and 8 with benign adrenal adenomas. We included patients aged 18 years or older who underwent adrenalectomy at SNUH between January 2000 and December 2019 and with available FFPE tumor tissues samples. Patients consented to donate tumor tissues, and their medical records were collected. Among the obtained tissues, those that had metastasized to sites other than the adrenal gland were not included in the analysis. In addition, cancers that metastasized to the adrenal gland, other than primary ACC, were excluded. Proteomics analysis was performed on the adrenal gland tissues of patients included in this cohort, and proteins related to the prognosis of ACC were identified by comparing benign adrenal adenoma and ACC, stages 1–2 and 3–4, ACCs, and stages 1–3 and 4 ACCs, respectively.

To validate the prognostic protein marker of ACC, we used the mRNA expression data from TCGA Pancancer Atlas database of the National Cancer Institute (https://portal.gdc.cancer.gov, accessed on 8 April 2021). The gene expression profile and clinical information of patients with ACC were downloaded from the TCGA database. The proteins associated with the prognosis of ACC were analyzed using TCGA data to validate the prognostic effect of the proteins.

### 2.2. Sample Preparation for Proteomics Analysis

Sample preparation for proteomic analysis of FFPE samples was performed as previously described [22,23]. Briefly, FFPE sections were incubated twice in xylene for 5 min, followed by 100% (*v*/*v*) ethanol twice for 3 min. The sections were then hydrated twice in 85% (*v*/*v*) ethanol for 1.5 min and distilled water for 5 min. Extraction buffer (4% SDS, 1 mM TCEP, and 0.3 M Tris pH 8.0) was added to deparaffinized tissue samples. After sonication, the samples were incubated at 95 °C for 2 h to ensure the most efficient de-crosslinking. The extracted proteins were precipitated by adding chilled acetone at a buffer to acetone volume ratio of 1:5, followed by incubation at −20 °C for 16 h. The protein (50 μg per sample) was digested following the filter-aided sample preparation procedure as previously described [22]. All peptides were acidified with 10% trifluoroacetic acid (TFA; Thermo Fisher Scientific, Waltham, MA, USA). Then, a three-fractionation strategy was applied to increase the proteome depth. The acidified peptides were loaded in homemade styrenedivinylbenzene reversed-phase sulfonate StageTips (3M, St. Paul, MN, USA), following previously described procedures [24]. Briefly, the peptides were washed three times with 100 μL of 0.2% TFA and sequentially eluted with three elution buffers with gradually increasing ACN concentration. The eluate was vacuum-centrifuged to dryness and stored at −80 °C.

### 2.3. LC–MS/MS and MS Data Analysis

All LC–MS/MS analyses were performed using Quadrupole Orbitrap mass spectrometers, Q-Exactive HF-X (Thermo Fisher Scientific, Waltham, MA, USA) coupled to an Ultimate 3000 RSLC system (Dionex, Sunnyvale, CA, USA) via a nanoelectrospray source, as described previously with some modifications [24,25]. The peptide samples were separated on a two-column setup with a trap column (PepMap 100, 0.3 mm I.D × 5 mm, C18 5 μm, Thermo Fisher Scientific, Waltham, MA, USA) and an analytical column (EASY-Spray™ PepMap RSLC C18, 50 µm I.D. × 50 cm, C18 1.9 µm, 100 Å, Thermo Fisher Scientific, Waltham, MA, USA) with a 90 min gradient from 8% to 26% acetonitrile at 300 nL/min and analyzed by mass spectrometry. The column temperature was maintained at 60 °C using a column heater. MaxQuant.Live (version 1.2; Max-Planck-Institute of Biochemistry, Planegg, Germany) was used to perform data-dependent acquisition (DDA) [26]. Survey scans (350 to 1650 *m*/*z*) were acquired with a resolution of 70,000 at *m*/*z* 200. A top 15 method was used to select the precursor ions with an isolation window of 1.2 *m*/*z*. The MS/MS spectra were acquired at an HCD-normalized collision energy of 28, with a resolution of 17,500, at *m*/*z* 200. The maximum ion injection times for the full scan and MS/MS scans were 20 and 100 ms, respectively.

Mass spectra were processed using MaxQuant version 1.6.1.0 [27]. The MS/MS spectra were searched against the Human Uniprot protein sequence database (December 2014, 88,657 entries) using the Andromeda search engine [28]. Primary searches were performed using a 6 ppm precursor ion tolerance to analyze the total protein levels. The MS/MS ion tolerance was set at 20 ppm. Cysteine carbamide methylation was set as a fixed modification. n-terminal acetylation of proteins and oxidation of methionine were set as variable modifications. Enzyme specificity was set to complete tryptic digestion. Peptides with a minimum length of six amino acids and up to two missed cleavages were considered. The minimal scores for unmodified and modified peptides were 0 and 40, respectively. The minimal delta scores for unmodified and modified peptides were 0 and 6, respectively. The minimum of unique and razor peptides for identification was set to 1. The peptide identifications across different LC-MS runs and the spectral library were matched by enabling the “match between runs” feature in MaxQuant. The required false discovery rate (FDR) was set to 1% at the peptide, protein, and modification levels. The mass spectrometry proteomics data were deposited to the ProteomeXchange Consortium via the PRIDE [29] partner repository with the dataset identifier PXD027404.

### 2.4. Label-Free Quantification and Bioinformatics Analysis

For label-free quantification, the intensity-based absolute quantification (iBAQ) algorithm [30] was used as part of the MaxQuant platform. Pair-wise comparisons were performed using the Perseus software [31]. For quantitative analysis of iBAQ data, we performed log_2_-transformation of iBAQ values. After filtering out proteins with at least 70% valid values in each group, missing values were imputed, assuming a normal distribution of 0.5 widths and 1.8 downshifts to simulate signals of low-abundance proteins. Finally, the data were normalized using the width adjustment function in the Perseus software. Student’s *t*-tests were performed for pair-wise comparisons of proteomes to detect DEPs with significant filtering criteria (*p*-value < 0.05, |fold-change| > 2).

To determine the important features, we employed three different algorithms: Relief [32], Information Gain [33], and analysis of variance (ANOVA) F-value [34] using the open-source software ORANGE (version 3.26; University of Ljubljana, Ljubljana, Slovenia). These feature selection models are well known for identifying features with good classification performance [35,36].

Canonical pathways, diseases, and functions were evaluated by Ingenuity Pathway Analysis (IPA, QIAGEN, Hilden, Germany) based on the annotated DEPs with matched gene names. The analytical algorithms embedded in IPA used the lists of DEPs to predict the biological processes and pathways. The statistical significance of both the gene ontology classification and enrichment analysis was determined using Fisher’s exact test. All statistical tests were two-sided, and a *p*-value of <0.05 was considered statistically significant.

### 2.5. Immunohistochemistry Staining

The protein identified as a prognostic protein marker of ACC in this study was verified by immunohistochemistry (IHC) staining in ACC and benign adrenal adenoma tissue. IHC was performed using anti-hnRNP A1 antibody (mouse monoclonal [9H10] to hnRNP A1 diluted in 1:100, Anti-hnRNP A1 antibody [9H10], AB5832, Abcam, Cambridge, UK), Ventana BenchMark XT Staining Systems, and OptiView DAB IHC Detection Kit (Ventana, Oro Valley, AZ, USA; #760-700). The 4 μm FFPE sections were dried at 60 °C for 1 h and deparaffinized by treatment with EZ Prep (Ventana #950-102) at 76 °C for 4 min. Antigen retrieval was performed by treatment with pH 8.4 Cell Conditioning 1 (CC1, Ventana #950-124) at 100 °C for 24 min. OptiView Peroxidase Inhibitor (3% hydrogen peroxide; Ventana) was treated at 37 °C for 4 min, and primary antibody was treated at 37 °C for 16 min. After treatment at 37 °C in OptiView HQ Universal Linker (Ventana) for 8 min, OptiView HRP Multimer (Ventana) at 37 °C for 8 min, and in OptiView DAB (Ventana) at 37 °C for 8 min. Hematoxylin (Ventana #760-2021) was treated at 37 °C for 8 min for counterstaining, and Bluing Reagent (Ventana #760-2037) was treated for 4 min at 37 °C for post counterstaining, followed by drying and mounting. The antibody dilution ratio was 1:100. IHC scoring was performed by a blinded pathologist (K.C.J.). The degree of nuclear staining was scored on an ordinal scale: 0, 1+, 2+, and 3+. The percentage of cells with nuclear staining was determined by visual assessment. The final IHC score was calculated using the following formula: 1 × (% of 1+ cells) + 2 × (% of 2+ cells) + 3 × (% of 3+ cells); the expression level ranged from 0 to 300.

### 2.6. Statistical Analysis

The baseline characteristics were presented as mean ± standard deviation for continuous variables and as number (%) for categorical variables. Kaplan–Meier analyses with log-rank tests and Cox proportional hazard regression models were performed for survival analyses of DEPs from feature selection. The above-median DEP expression levels were used for survival analyses. For multivariate Cox regression analyses, age per 10-year increment and clinical stages 1–2, 3, and 4 were used due to the small number of samples with stages 1 and 2. To evaluate the improvements in predicting the performance of significant DEPs, the C-index and category-free net reclassification index (NRI) were calculated, and the age and staging models were used as the reference. Statistical significance was set at *p*-value < 0.05. Statistical analyses were performed using R (version 3.6.1; R Foundation; Vienna, Austria; https://www.r-project.org, accessed on 8 April 2021).

### 2.7. Ethical Statement

This study was conducted in accordance with the principles of the Declaration of Helsinki and was reviewed and approved by the Institutional Review Board (IRB) of SNUH (IRB no. H-1912-139-1091). The requirement for obtaining patient consent was waived due to the retrospective nature of the study, and analyses were performed using de-identified data. This study was also conducted following TCGA Human Subject Protection and Data Access Policies.

## 3. Results

### 3.1. Baseline Characteristics of the Study Participants

The baseline characteristics of 37 ACC and 8 benign patients are shown in Table 1. The mean age of patients with ACC was 48.5 ± 12.9 years, and 40.5% were men. The mean age of benign patients was 51.9 ± 10.5 years, and 50.0% were men.

Approximately 5.4% of ACC patients had stage 1 disease, 18.9% had stage 2 disease, 45.9% had stage 3 disease, and 29.7% had stage 4 disease. The average follow-up duration was 4.0 (1.2–8.3) years, and 51.4% died during the follow-up period. Among patients with ACC, 48.6% presented hypercortisolism, and 18.9% and 29.7% exhibited a mitotic count of 20/HPF or higher and a Ki67 index of 20% or higher, respectively.

### 3.2. Results of Proteomic Analysis

To find the prognostic biomarker of ACC, MS-based label-free quantification was performed using a formalin-fixed paraffin-embedded tissue of the primary adrenal gland tumor obtained through adrenalectomy (Figure 1A). In combination with a matching library consisting of FFPE pooling samples as well as ACC frozen tissues, a total of 8261 proteins and 95,541 unique peptides were identified with a false discovery rate (FDR) <1% at PSM and protein level (Appendix A). Among them, 7812 proteins and 7669 proteins were identified and quantified in the individual samples, respectively. On average, 5563 proteins and 24,867 unique peptides were quantified in each independent sample (Appendix A).

A total of 6908 common proteins were identified in both benign adrenal adenomas and ACCs. A total of 7156 proteins were commonly observed in stages 1–2 and 3–4 ACCs, while 7132 proteins were commonly observed in stages 1–3 and 4 ACCs (Figure 1B).

The signal intensities of all proteins were plotted along with several well-known markers of ACC (Figure 1C).

### 3.3. Analysis of Differentially Expressed Proteins

Volcano plots were presented to show DEPs. The differences in the protein expression levels of benign adrenal adenomas and ACC and the stages of ACC were compared using Student’s *t*-test (*p* < 0.05; fold-change ≥ 2.0) (Figure 2). Compared with benign adrenal adenomas, 452 proteins were downregulated, and 554 proteins were upregulated in ACC. Compared to stages 1–2 ACCs, 222 proteins were downregulated, and 317 proteins were upregulated in stages 3–4 ACCs. Moreover, 243 proteins were downregulated, and 168 proteins were upregulated in stage 4 ACCs compared with stages 1–3 ACCs.

To identify proteomic signatures with diagnostic power, we selected TOP20 featured DEPs using feature selection algorithms including ReliefF, infoGain, and ANOVA (Table 2). DEPs with high rankings in all algorithms are displayed as overlapping circles in Figure 3. We assessed the signal intensity of high-ranked DEPs in two or more feature selection methods (Appendix A).

### 3.4. Ingenuity Pathway Analysis (IPA)

We performed IPA using significant DEPs to identify the canonical pathways in ACCs. Pathways with a *p*-value of 0.05 or less between the two groups are shown in Appendix A and the top 10 pathways are shown in Figure 4. When comparing benign adrenal adenomas and ACC, glioma signaling and sirtuin signaling pathways were significantly activated in the ACC. When comparing stages 1–2 and 3–4 ACCs, acute phase response signaling, LXR/RXR activation, production of nitric oxide and reactive oxygen species in macrophages, GP6 signaling pathway, and glioma invasiveness signaling were upregulated in stages 3–4 ACCs. When comparing stages 1–3 and 4 ACCs, the GP6 signaling pathway and leukocyte extravasation signaling were activated in stage 4 ACCs.

### 3.5. Selection of Prognostic Protein Biomarkers in the Seoul National University Hospital (SNUH) Cohort

Survival analyses were performed in the SNUH cohort to assess whether featured DEPs affected patients’ survival. Among DEPs, 117 proteins, which are the top 20 proteins of the three algorithms for feature selection (ReliefF, infoGain, and ANOVA), were analyzed (benign adrenal adenoma vs. ACC, 39 proteins; stages 1–2 ACCs vs. stages 3–4 ACCs, 41 proteins; stages 1–3 ACC vs. stage 4 ACCs, 37 proteins). Each DEP was divided into two groups based on the median values. In the log-rank test, 9 DEPs of the 117 DEPs were significantly related to patients’ survival (Figure 5). In the Cox proportional hazard regression models, heterogeneous nuclear ribonucleoprotein A1 (HNRNPA1), C8A, CHMP6, latent transforming growth factor β-binding protein 4 (LTBP4), SPR, NCEH1, mitochondrial ribosomal protein S23 (MRPS23), polymerase delta interacting protein 2 (POLDIP2), and Williams–Beuren syndrome chromosome region 16 (WBSCR16) were significantly associated with the prognosis of ACC (Figure 6). Patients with high expression of HNRNPA1, C8A, CHMP6, LTBP4, and NCEH1 were at higher risk of mortality (hazard ratios (HR; 95% confidence interval [CIs]): 2.84 (1.07–7.53), 3.52 (1.28–9.63), 2.83 (1.06–7.56), 2.63 (1.03–6.73), and 2.98 (1.10–8.05), respectively). By contrast, patients with high expression of SPR, MRPS23, POLDIP2, and WBSCR16 had a lower risk of mortality (HRs (95% CIs): 0.36 (0.14–0.97), 0.30 (0.11–0.79), 0.30 (0.12–0.81), and 0.35 (0.13–0.92), respectively). In age- and stage-adjusted Cox proportional hazard regression models, the expression of HNRNPA1, C8A, CHMP6, MRPS23, and WBSCR16 remained significant. In addition, when calculating each C-index of nine proteins added to age and stage, all of them showed a higher C-index compared with age and stage alone. However, when the net reclassification index (NRI) was calculated, only HNRNPA1, MRPS23, and WBSCR16 exhibited a significantly higher C-index compared with age and stage alone (Table 3).

### 3.6. Validation of the Prognostic Value of Candidate Protein Biomarkers in the TCGA Cohort

We validated nine candidate protein biomarkers derived from the SNUH cohort in the TCGA cohort using Cox proportional hazard regression analyses. The baseline characteristics of 78 patients with ACC in the TCGA study are presented in Appendix A; no significant differences were observed in age, sex, and follow-up duration between the SNUH cohort and TCGA cohort (Appendix A). However, a significant difference was found in the distribution of the initial stages between the two groups. In the log-rank test and univariate Cox proportional hazard regression models, HNRNPA1, LTBP4, MRPS23, POLDIP2, and WBSCR16 were significantly associated with mortality (Figure 7 and Figure 8). However, in the multivariate analyses after adjusting for age and stage, only HNRNPA1 remained significant (Figure 8).

We further compared the prognostic values of candidate protein biomarkers using the C-index and NRI (Table 3). In the TCGA cohort, the C-index and NRI of HNRNPA1, LTBP4, MRPS23, POLDIP2, and WBSCR16 were significantly higher than those of age and stage alone. In addition, the combination of HNRNPA1, MRPS23, and WBSCR16 significantly predicted the prognosis in the TCGA cohort.

### 3.7. Validation of the Prognostic Candidate Protein Biomarker by Immunohistochemistry Staining

Among the five proteins, HNRNPA1, LTBP4, MRPS23, POLDIP2, and WBSCR16, which were confirmed to be significantly correlated with ACC survival in the TCGA cohort, IHC staining was performed with HNRNPA1, which was also associated with poor prognosis in other tumors (Appendix A). A positive correlation was observed between HNRNPA1 intensity obtained by MS data analysis and IHC staining intensity (*r* = 0.478, *p* = 0.029, Appendix A). Moreover, when the tissues were divided into two groups, low and high, according to the expression level of HNRNP1A, there was a significant difference in IHC score between the two groups (*p* = 0.002, Appendix A).

## 4. Discussion

In the present study, we identified the prognostic protein biomarkers of ACCs using LC–MS/MS of FFPE tumor tissues. To date, only transcriptome analysis has been performed on ACCs [13,37,38,39]. Despite the comprehensive permitting analysis of mRNA transcripts, these studies cannot demonstrate whether the observed modulation in mRNA level corresponds to a consequent modulation in protein levels. Indeed, transcript abundance at a steady state only partially predicts the protein levels in various systems [40]. Thus, a comprehensive analysis of the global proteome of ACCs not only contributes to a better understanding of the biology of carcinomas but also defines new targets for therapy and even practical biomarkers for the early detection of disease. However, few proteomic studies have been performed on ACCs [9,41,42]. To the best of our knowledge, this is the first in-depth proteomic analysis of ACCs. We provided an in-depth comparison of the ACC proteomes at different stages and identified a total of 7000 individual proteins across disease stages.

In the SNUH cohort, we first identified DEPs using a combination of three machine learning algorithms and then determined the final candidate protein biomarkers, HNRNPA1, LTBP4, MRPS23, POLDIP2, and WBSCR16, through survival analyses. In the TCGA cohort, we confirmed the prognostic value of the five candidate proteins. Among them, HNRNPA1 had the most potent prognostic value for survival. Furthermore, even when IHC staining was performed, HNRNPA1 was observed to be associated with the prognosis of ACC.

HNRNPA1 is a family of RNA-binding proteins. HNRNPA1 is involved in gene expression and signal transduction by performing various functions such as processing heterogeneous nuclear RNAs into mature mRNAs, RNA splicing, transactivation of gene expression, and modulation of protein translation [43]. To date, the role of HNRNPA1 in ACC has not been studied, but other studies have shown that overexpression of HNRNPA1 in hepatocellular carcinoma and gastric cancer promotes tumor invasion and is related to poor prognosis [44,45]. HNRNPA1 is involved in the alternative splicing of the insulin receptor gene [46]. The insulin receptor is also a receptor of insulin-like growth factor 2, which is involved in ACC tumorigenesis [47].

Moreover, higher expression of LTBP4 was associated with a poor prognosis of ACCs, although the role of LTBP4 in ACC has not been studied so far. LTBP4 is one of a family of latent TGF-β binding proteins involved in the correct folding and secretion of TGF-β [48]. The direct role of the TGF-β signaling pathway in the tumorigenesis of ACCs has not yet been reported. However, the TGF-β signaling pathway in premalignant cells suppresses proliferation and promotes apoptosis; in late-stage cancers, it provokes epithelial-to-mesenchymal transition and metastasis [49]. Previous studies regarding the relationship between LTBP4 and tumorigenesis have reported contradictory results. Knockdown of LTBP4 in mice was associated with the development of epithelial carcinoma [50], and the expression of LTBP4 was lower in esophageal cancer and mammary carcinoma than in normal tissues [51,52]. However, given the dual role of the TGF-β signaling pathway in early and late-stage cancers, the role of LTBP4 in ACC tumorigenesis and metastasis requires further validation.

MRPS23 was a favorable prognostic marker in our study. It is a 28S subunit protein encoded by a nuclear gene that plays a role in mitochondrial protein translation [53]. However, the high expression of MRPS23 was associated with tumor proliferation and angiogenesis by changing the hypoxic state of tumor cells and increasing mitochondrial activity [54]. Therefore, previous studies reported the following findings: the abnormal expression of MRPS23 might serve as a poor prognostic factor of the tumor size in hepatocellular carcinomas [53], breast cancers [55], and cervical cancer [54]. However, in our study, MRPS23 was downregulated in metastatic ACCs compared with that in non-metastatic ACCs. Thus, the role of MRPS23 in ACC metastasis might be different from that in tumor proliferation.

Moreover, the high expression of POLDIP2 was associated with low mortality risk. POLDIP2, also called polymerase delta-interacting protein of 38 kDa, is responsible for DNA replication and repair, mitochondrial function, and cell cycle regulation [56,57,58,59]. In a previous study on non-small cell lung carcinoma (NSCLC), POLDIP2 expression was lower in NSCLC tumor tissues, but the overexpression of POLDIP2 increased the growth and invasiveness of NSCLC cell lines [60,61]. POLDIP2 interacts with multiple protein partners and participates in numerous cellular processes. Thus, the actual role of POLDIP2 in tumorigenesis and metastasis remains unknown.

WBSCR16 is a guanine nucleotide exchange factor that plays an important role in mitochondrial fusion [62]. Mitochondrial fusion is pivotal for maintaining cellular homeostasis and the proper intracellular distribution of organelles [63]. Although the role of this protein in cancer cells has not been studied so far, mitochondrial dynamics have emerged as new therapeutic avenues for targeting cancer stem cells [63]. Thus, WBSCR16 may play a role in ACC metastasis.

Several molecular markers were suggested to be prognostic factors [3]. The combination of budding uninhibited by benzimidazoles 1 homolog beta and PTEN-induced putative kinase 1 is a strong predictor of recurrence and overall survival [64,65]. The histone methyltransferase EZH2 results in deregulated P53/RB/E2F pathway activity and is associated with a poorer prognosis in patients with ACC [66]. The abundance of VAV2, which is a guanine nucleotide exchange factor for small GTPases and is induced by steroidogenic factor-1, predicted the patient’s overall survival independent of age, tumor stage, and Ki-67 index [67,68]. Low DAXX and high phospho-mTOR expression levels are associated with a poor prognosis of ACC [69]. PTTG1, associated with cancer invasiveness, can also be a diagnostic and prognostic marker for ACC [10]. Recently, pan-genomic studies have reported different survival outcomes by dividing the ACC into different molecular subgroups [13,14,15]. However, these studies were related to genetic alterations and mRNA expression, which limits the application of genetic studies to actual clinical practice. Tian et al. constructed prognostic models derived from RPPA-based proteins and gene expression profiling and validated the models using immunohistochemistry [70]. They suggested that higher fatty acid synthase, fibronectin, transferrin receptor 1, and TSC complex subunit 1 expression indicated worse overall survival in patients with ACC [70]. Our candidate protein biomarkers, including HNRNPA1, LTBP4, MRPS23, POLDIP2, and WBSCR16, were suggested, and the mechanisms of action of each protein in ACC tumorigenesis or metastasis should be elucidated.

However, this study has several limitations. First, due to the rarity of ACCs, the small sample size may not be enough to discover and validate protein biomarkers. Therefore, our results should be confirmed in independent patients with ACC. Although we further validated our results in the TCGA cohort, we used TCGA data from RNA sequencing because of the lack of protein numbers in RPPA data. We analyzed the expression of candidate protein biomarkers in terms of overall survival. Therefore, the prognostic value of each protein biomarker for risk of recurrence was not included.

## 5. Conclusions

The present study identified HNRNPA1, LTBP4, MRPS23, POLDIP2, and WBSCR16 as protein biomarkers to predict the prognosis of ACC. These biomarkers might guide clinicians in determining the adjuvant treatment after surgery and improve the prognosis of patients with ACC. Moreover, these proteins may be potential therapeutic targets in patients with ACC. Further functional studies and external validation studies are required to identify the mechanism of action and the generalization of new protein biomarkers.

## Figures and Tables

**Figure 1 cancers-13-03890-f001:**
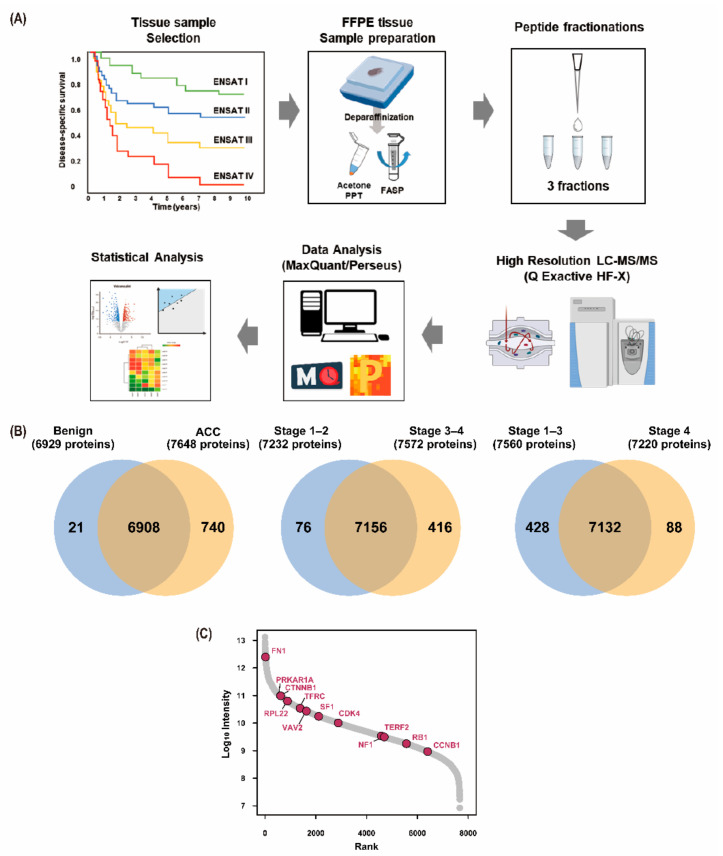
Overall workflow and results of proteomic analysis. (**A**) Overall workflow for proteomic analysis of adrenal cortical carcinoma FFPE tissues with respect to ENSAT staging is presented. This figure was created with Biorender.com (accessed on 10 June 2021) and exported under a paid subscription; (**B**) The number of total proteins expressed in benign adrenal adenomas and ACC, stages 1–2 and 3–4 ACCs, and stages 1–3 and 4 ACCs are expressed, respectively. The overlapping part of the circle indicates the number of proteins expressed in both groups; (**C**) The intensity of the expression of well-known proteins in the ACC. ACC—adrenal cortical carcinoma; FASP—filter aided sample preparation; FFPE—formalin-fixed paraffin-embedded; ENSAT—European Network for the Study of Adrenal Tumors; LS-MS/MS—liquid chromatography–tandem mass spectrometry; PPT—precipitation.

**Figure 2 cancers-13-03890-f002:**
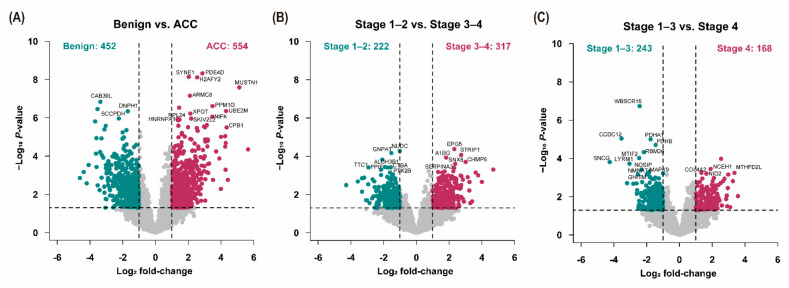
Volcano plots for differential expression of proteins. Volcano plots showed significantly differentially abundant proteins. The −log_10_ (*p*-value) is plotted against log_2_ (fold-change). The non-axial vertical lines denote a two-fold difference in expression, while the non-axial horizontal line denotes *p* = 0.05, which is our significance threshold. The differential expression of proteins in (**A**) benign adrenal adenomas and ACC, (**B**) stages 1–2 and 3–4 ACCs, and (**C**) stages 1–3 and 4 ACCs is expressed, respectively. ACC—adrenal cortical carcinoma.

**Figure 3 cancers-13-03890-f003:**
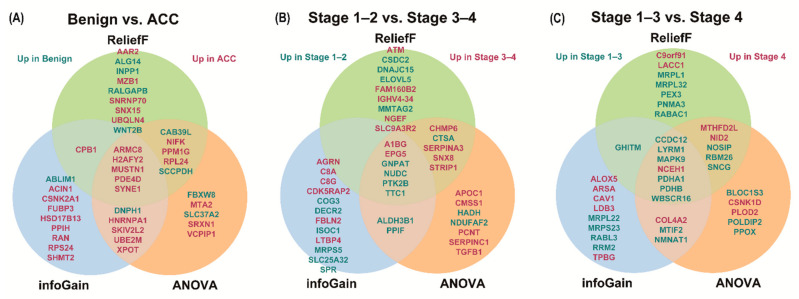
Feature selection of proteins for classification (top 20 DEPs). Feature selection was performed using ReliefF, infoGain, and ANOVA for DEP. Using each algorithm, the top 20 DEPs were obtained, and proteins with high rankings in all algorithms are displayed in overlapping circles. Differential expressions of proteins in (**A**) benign adrenal adenomas and ACC, (**B**) stages 1–2 and 3–4 ACCs, and (**C**) stages 1–3 and 4 ACCs are expressed, respectively. Red letters indicate highly expressed proteins in the worse group, while green letters indicate highly expressed proteins in the better group. ACC—adrenal cortical carcinoma; ANOVA—analysis of variance; DEP—differential expression of proteins.

**Figure 4 cancers-13-03890-f004:**
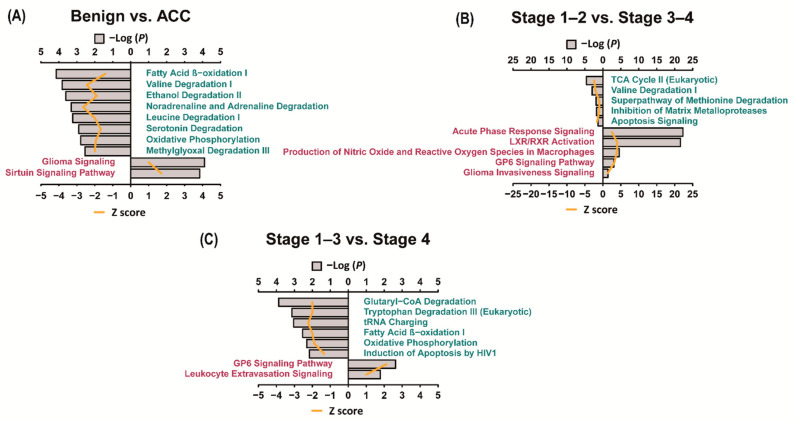
IPA analyses of the differential expression of proteins (Canonical pathway). IPA analysis was performed in the differentially expressed proteins identified in each group, and canonical pathways with low *p*-values were presented. (**A**) Pathways identified in benign adrenal adenomas are indicated in green, while pathways identified in ACC are indicated in red; (**B**) Pathways identified in stages 1–2 ACC are indicated in green, while pathways identified in stages 3–4 are indicated in red; (**C**) Pathways identified in stages 1–3 are indicated in green, while pathways identified in stage 4 are indicated in red. The Z-score indicates that the level of protein expression was significantly high, and it was significant when the absolute value was more than 1. ACC—adrenal cortical carcinoma; GP6—Glycoprotein VI; IPA—ingenuity pathway analysis; LXR/RXR—Liver X Receptor-Retinoid X Receptor; TCA—tricarboxylic acid cycle.

**Figure 5 cancers-13-03890-f005:**
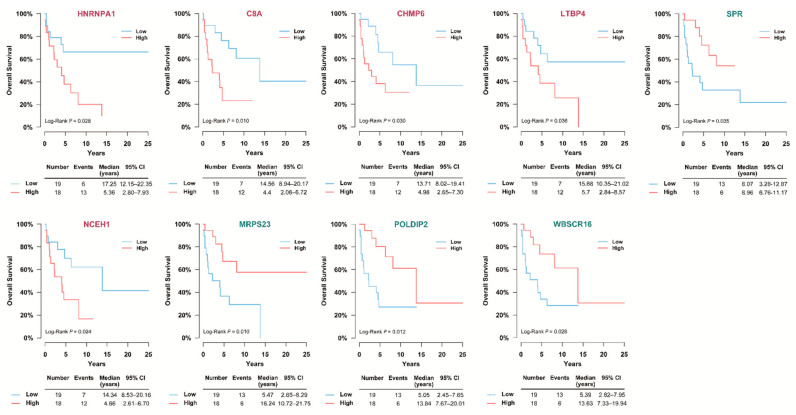
Kaplan–Meier analyses of significant DEPs in SNUH cohort. When Kaplan–Meier analyses were performed on the featured proteins, nine proteins with significant differences were identified. It was analyzed by dividing it into low and high groups based on the median value. When comparing benign adrenal adenomas with ACC, the high expression of HNRNPA1 was associated with poor overall survival. When comparing stage 1–2 and stage 3–4 ACCs, high expressions of C8A, CHMP6, and LTBP4 and low expression of SPR were associated with poor overall survival. In addition, when comparing stages 1–3 and stage 4 ACCs, high expression of NCEH1 and low expressions of MRPS23, POLDIP2, and WBSCR16 were associated with poor overall survival. ACC—adrenal cortical carcinoma; C8A—complement C8 alpha chain; CHMP6—charged multivesicular body protein 6; CI—confidence interval; DEP—differential expression of proteins; HNRNPA1—heterogeneous nuclear ribonucleoprotein A1; LTBP4—latent transforming growth factor beta binding protein 4; MRPS23—mitochondrial ribosomal protein S23; NCEH1—neutral cholesterol ester hydrolase 1; POLDIP2—polymerase delta-interacting protein 2; SNUH—Seoul National University Hospital; SPR—sepiapterin reductase; WBSCR16—Williams-Beuren syndrome chromosome region 16.

**Figure 6 cancers-13-03890-f006:**
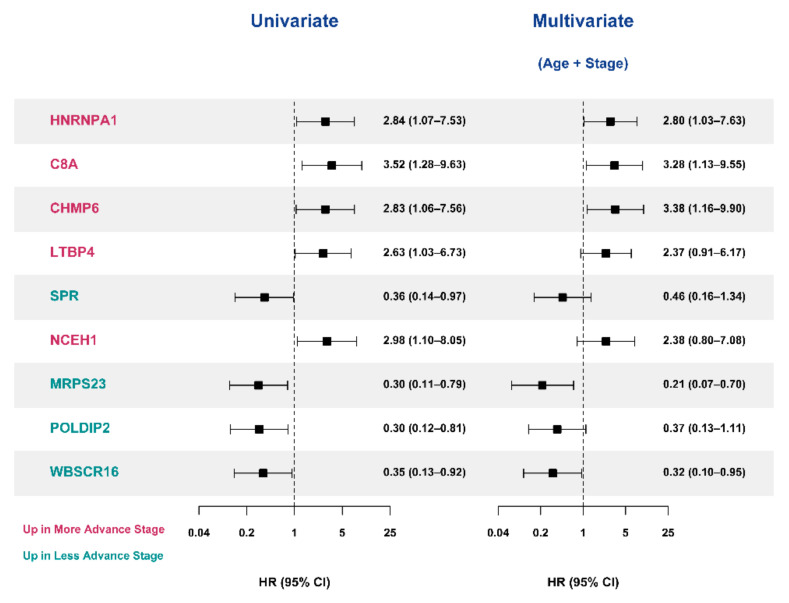
Cox proportional hazard regression models of significant DEPs in SNUH cohort. When Cox proportional hazard regression model analysis was performed on the feature-selected protein, nine proteins with significant differences were identified. ACC—adrenal cortical carcinoma; C8A—complement C8 alpha chain; CHMP6—charged multivesicular body protein 6; CI—confidence interval; DEP—differential expression of proteins; HNRNPA1—heterogeneous nuclear ribonucleoprotein A1; LTBP4—latent transforming growth factor beta binding protein 4; MRPS23—mitochondrial ribosomal protein S23; NCEH1—neutral cholesterol ester hydrolase 1; POLDIP2—polymerase delta-interacting protein 2; SNUH—Seoul National University Hospital; SPR—sepiapterin reductase; WBSCR16—Williams-Beuren syndrome chromosome region 16.

**Figure 7 cancers-13-03890-f007:**
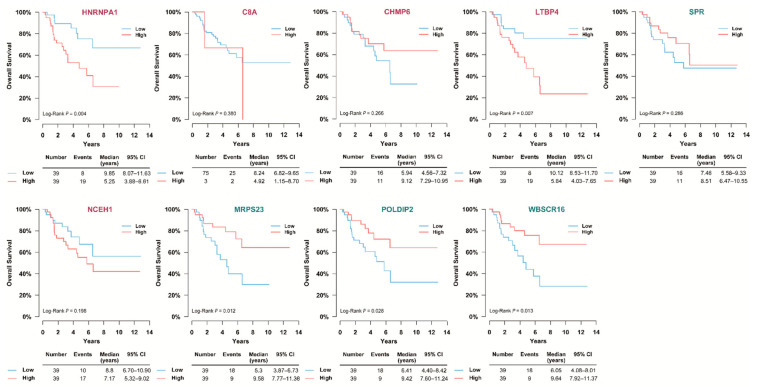
Kaplan–Meier analyses of significant DEPs in TCGA study. We tried to validate DEPs, which showed a significant difference in survival in the SNUH cohort, by performing Kaplan–Meier analyses in the TCGA study. It was analyzed by dividing it into low and high groups based on the median value. C8A—complement C8 alpha chain; CHMP6—charged multivesicular body protein 6; CI—confidence interval; DEP—differential expression of proteins; HNRNPA1—heterogeneous nuclear ribonucleo-protein A1; LTBP4—latent transforming growth factor beta binding protein 4; MRPS23—mitochondrial ribosomal protein S23; NCEH1—neutral cholesterol ester hydrolase 1; POLDIP2—polymerase delta-interacting protein 2; SPR—sepiapterin reductase; TCGA—The Cancer Genome Atlas; WBSCR16—Williams-Beuren syndrome chromosome region 16.

**Figure 8 cancers-13-03890-f008:**
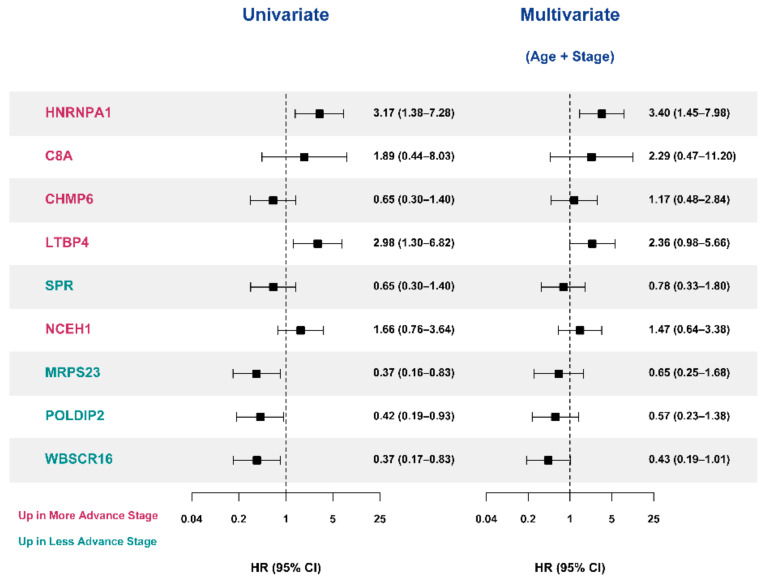
Cox proportional hazard regression models of significant DEPs in TCGA study. Cox proportional hazard regression model analysis was performed on DEPs, which showed a significant difference in survival in the TCGA study. C8A—complement C8 alpha chain; CHMP6—charged multivesicular body protein 6; CI—confidence interval; DEP—differential expression of proteins; HNRNPA1—heterogeneous nuclear ribonucleo-protein A1; LTBP4—latent transforming growth factor beta binding protein 4; MRPS23—mitochondrial ribosomal protein S23; NCEH1—neutral cholesterol ester hydrolase 1; POLDIP2—polymerase delta-interacting protein 2; SNUH—Seoul National University Hospital; SPR—sepiapterin reductase; TCGA—The Cancer Genome Atlas; WBSCR16—Williams-Beuren syndrome chromosome region 16.

**Table 1 cancers-13-03890-t001:** Baseline characteristics of patients with adrenal cortical carcinoma (*n* = 37) and benign adrenal adenoma (*n* = 8).

Variable	ACC (*n* = 37)	Benign (*n* = 8)
Age	48.5 ± 12.9	51.9 ± 10.5
Male	15 (40.5)	4 (50.0)
Initial Stage (ENSAT)		
I	2 (5.4)	-
II	7 (18.9)	-
III	17 (45.9)	-
IV	11 (29.7)	-
Death	19 (51.4)	-
Follow-up, years (IQR)	4.0 (1.3–8.1)	-
Cortisol Secretion ^a^		
Yes	18 (48.6)	-
No	9 (24.3)	-
Mitosis Count ^b^		
≥20/HPF	7 (18.9)	-
<20/HPF	22 (59.5)	-
Ki67 ^c^		
≥20%	11 (29.7)	-
10–19%	4 (10.8)	-
<10%	7 (18.9)	-

Values are expressed as mean ± standard deviation, number (%), or median (IQR). ^a^ The available number was 27 because of missing values; ^b^ The available number was 29 because of missing values; ^c^ The available number was 22 because of missing values. ACC—adrenal cortical carcinoma; ENSAT—European Network for the Study of Adrenal Tumor; HPF—high power field; IQR—interquartile range.

**Table 2 cancers-13-03890-t002:** Top 20 proteins of machine learning analysis.

Rank	Benign vs. ACC						Stage 1–2 vs. 3–4						Stage 1–3 vs. 4					
	InfoGain		ANOVA		ReliefF		InfoGain		ANOVA		ReliefF		InfoGain		ANOVA		ReliefF	
	Gene Name	Score	Gene Name	Score	Gene Name	Score	Gene Name	Score	Gene Name	Score	Gene Name	Score	Gene Name	Score	Gene Name	Score	Gene Name	Score
1	*H2AFY2*	0.787	*PDE4D*	62.944	*MUSTN1*	0.321	*EPG5*	0.681	*EPG5*	24.961	*NUDC*	0.239	*WBSCR16*	0.658	*WBSCR16*	52.070	*WBSCR16*	0.303
2	*ABLIM1*	0.654	*SYNE1*	60.441	*ARMC8*	0.298	*GNPAT*	0.531	*NUDC*	24.001	*GNPAT*	0.226	*CCDC12*	0.658	*CCDC12*	31.394	*NOSIP*	0.234
3	*DNPH1*	0.654	*H2AFY2*	60.114	*SNX15*	0.296	*DECR2*	0.531	*GNPAT*	23.111	*FAM160B2*	0.217	*PDHA1*	0.658	*PDHA1*	31.002	*CCDC12*	0.196
4	*ACIN1*	0.531	*MUSTN1*	53.536	*SYNE1*	0.291	*PPIF*	0.531	*STRIP1*	22.252	*SNX8*	0.191	*LYRM1*	0.581	*PDHB*	28.034	*GHITM*	0.188
5	*RPS24*	0.531	*ARMC8*	48.449	*PDE4D*	0.276	*SPR*	0.531	*A1BG*	21.073	*NGEF*	0.182	*RRM2*	0.581	*RBM26*	24.492	*MRPL32*	0.184
6	*FUBP3*	0.526	*CAB39L*	44.931	*CAB39L*	0.275	*SLC25A32*	0.531	*NDUFAF2*	20.221	*CTSA*	0.170	*NCEH1*	0.519	*MTIF2*	21.829	*PDHB*	0.179
7	*SYNE1*	0.526	*PPM1G*	42.526	*RALGAPB*	0.270	*COG3*	0.531	*CHMP6*	19.245	*A1BG*	0.166	*ARSA*	0.519	*CSNK1D*	21.478	*SNCG*	0.171
8	*ARMC8*	0.526	*MTA2*	41.666	*SNRNP70*	0.251	*ISOC1*	0.531	*SNX8*	18.413	*ATM*	0.164	*CAV1*	0.519	*SNCG*	20.043	*PDHA1*	0.170
9	*HSD17B13*	0.526	*SLC37A2*	40.937	*AAR2*	0.245	*TTC1*	0.522	*ALDH3B1*	17.218	*PTK2B*	0.163	*RABL3*	0.519	*LYRM1*	19.369	*RBM26*	0.167
10	*PDE4D*	0.526	*UBE2M*	39.877	*MZB1*	0.239	*MRPS5*	0.522	*CTSA*	17.069	*STRIP1*	0.160	*MRPS23*	0.510	*NCEH1*	17.250	*NCEH1*	0.153
11	*UBE2M*	0.526	*DNPH1*	39.713	*H2AFY2*	0.237	*AGRN*	0.478	*TTC1*	17.034	*CHMP6*	0.158	*TPBG*	0.499	*NOSIP*	17.024	*PNMA3*	0.152
12	*SKIV2L2*	0.526	*XPOT*	38.513	*NIFK*	0.230	*FBLN2*	0.478	*HADH*	17.000	*MMTAG2*	0.157	*PDHB*	0.475	*NMNAT1*	16.105	*LYRM1*	0.146
13	*SHMT2*	0.526	*NIFK*	36.911	*ALG14*	0.225	*CDK5RAP2*	0.478	*CMSS1*	16.775	*DNAJC15*	0.156	*MAPK9*	0.475	*COL4A2*	15.781	*NID2*	0.142
14	*CPB1*	0.526	*SCCPDH*	36.042	*CPB1*	0.220	*A1BG*	0.458	*APOC1*	16.197	*EPG5*	0.153	*MTIF2*	0.475	*MTHFD2L*	15.740	*RABAC1*	0.140
15	*XPOT*	0.526	*SKIV2L2*	35.957	*WNT2B*	0.210	*C8G*	0.458	*PPIF*	16.157	*TTC1*	0.153	*NMNAT1*	0.475	*POLDIP2*	15.577	*C9orf91*	0.138
16	*HNRNPA1*	0.526	*HNRNPA1*	35.727	*SCCPDH*	0.201	*LTBP4*	0.458	*PCNT*	15.578	*SERPINA3*	0.151	*MRPL22*	0.475	*MAPK9*	15.468	*MTHFD2L*	0.138
17	*CSNK2A1*	0.526	*VCPIP1*	35.452	*PPM1G*	0.200	*C8A*	0.458	*SERPINA3*	15.499	*IGHV4-34*	0.146	*LDB3*	0.475	*NID2*	15.440	*MAPK9*	0.137
18	*MUSTN1*	0.526	*RPL24*	35.172	*RPL24*	0.199	*NUDC*	0.396	*SERPINC1*	15.308	*CSDC2*	0.144	*ALOX5*	0.475	*BLOC1S3*	15.217	*MRPL1*	0.137
19	*PPIH*	0.526	*FBXW8*	34.574	*UBQLN4*	0.198	*PTK2B*	0.396	*PTK2B*	15.098	*ELOVL5*	0.144	*COL4A2*	0.431	*PPOX*	15.193	*LACC1*	0.136
20	*RAN*	0.526	*SRXN1*	32.834	*INPP1*	0.196	*ALDH3B1*	0.396	*TGFB1*	15.072	*SLC9A3R2*	0.142	*GHITM*	0.427	*PLOD2*	15.117	*PEX3*	0.135

ACC—adrenal cortical carcinoma; ANOVA—analysis of variance; InfoGain—Information Gain.

**Table 3 cancers-13-03890-t003:** C-index and net reclassification index.

Variables	SNUH Cohort			TCGA Cohort		
	C-Index (95% CI)	NRI (95% CI)	*p*-Value for NRI	C-Index (95% CI)	NRI (95% CI)	*p*-Value for NRI
Age+Stage	0.690 (0.551—0.829)	-	-	0.674 (0.552—0.796)	-	-
Age+Stage+HNRNPA1	0.704 (0.563—0.845)	0.813 (0.225—1.401)	**0.007**	0.757 (0.662—0.852)	0.625 (0.179—1.070)	**0.006**
Age+Stage C8A	0.727 (0.612—0.841)	0.152 (−0.480—0.784)	0.637	0.674 (0.550—0.798)	0.228 (−0.070—0.526)	0.134
Age+Stage+CHMP6	0.756 (0.645—0.867)	0.152 (−0.480—0.784)	0.637	0.683 (0.568—0.797)	−0.234 (−0.704—0.236)	0.329
Age+Stage+LTBP4	0.702 (0.561—0.842)	0.596 (−0.018—1.211)	0.057	0.752 (0.639—0.865)	0.585 (0.138—1.031)	**0.010**
Age+Stage+SPR	0.718 (0.583—0.852)	0.503 (−0.088—1.094)	0.095	0.681 (0.559—0.803)	0.311 (−0.154—0.776)	0.190
Age+Stage+NCEH1	0.695 (0.550—0.839)	0.485 (−0.140—1.111)	0.128	0.689 (0.575—0.803)	0.428 (−0.030—0.885)	0.067
Age+Stage+MRPS23	0.740 (0.618—0.862)	0.819 (0.239—1.398)	**0.006**	0.729 (0.632—0.825)	0.508 (0.052—0.963)	**0.029**
Age+Stage+POLDIP2	0.733 (0.612—0.855)	0.591 (−0.024—1.205)	0.060	0.741 (0.666—0.816)	0.508 (0.052—0.963)	**0.029**
Age+Stage+WBSCR16	0.740 (0.606—0.874)	0.702 (0.098—1.305)	**0.023**	0.743 (0.668—0.818)	0.508 (0.052—0.963)	**0.029**
Age+Stage+HNRNPA1+MRPS23+WBSCR16 ^a^	0.708 (0.579—0.838)	0.175 (−0.447—0.797)	0.58	0.752 (0.659—0.845)	0.492 (0.091—0.894)	**0.016**

Age per 10 years and stage of 1–2 vs. 3 vs. 4 is used. HNRNPA1, C8A, CHMP6, LTBP4, and NCEH1 were analyzed as high versus low expressions, and SPR, MRPS23, POLDIP2, and WBSCR16 were analyzed as high versus low expressions; ^a^ The analysis compared patients with high HNRNPA1, low MRPS23 and WBSCR16, and those who did not. Values in bold in the *p*-value indicate statistical significance. C8A—complement C8 alpha chain; CHMP6—charged multivesicular body protein 6; C-index—concordance index; CI—confidence interval; HNRNPA1—heterogeneous nuclear ribonucleo-protein A1; LTBP4—latent transforming growth factor beta binding protein 4; MRPS23—mitochondrial ribosomal protein S23; NCEH1—neutral cholesterol ester hydrolase 1; NRI—net reclassification improvement; POLDIP2—polymerase delta-interacting protein 2; SNUH—Seoul National University Hospital; SPR—sepiapterin reductase; TCGA—The Cancer Genome Atlas; WBSCR16—Williams-Beuren syndrome chromosome region 16.

## Data Availability

The data presented in this study are available on request from the corresponding author.

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
