# Peer review of "Mass Spectrometry-Based Proteomic Discovery of Prognostic Biomarkers in Adrenal Cortical Carcinoma"

_cancers, 2021, doi:10.3390/cancers13153890_

Round 1
Reviewer 1 Report
The manuscript of Jang et al. describes a proteome study on human adrenal cortical carcinoma, a tumor type with limited available molecular profiling data. While I really appreciate the quality of most parts of the data as well as the data interpretation strategy, I request a few important corrections before the manuscript may become acceptable for publication.
First of all and most importantly, no prognostic biomarkers are presented in this study, it is rather prognostic biomarker candidates. A prognostic biomarker has to be validated in a prognostic study, here we are dealing with retrospective data analysis only. Per se this is no problem, but the title, abstract, discussion and conclusions and several related statements need to be corrected accordingly. In line, a proteome study does not define new targets for therapy as stated in line 339, it may at best suggest new targets.
II) I am missing some details in the Materials and Methods section. Please specify the vendor of the analytical column. Did you reduce the data matrix based on number of positive identifications per group or include single identification events? How did you deal with missing values? It seems you did not consider methylation of lysines, characteristic for FFPE samples, what you actually should do. What was the number of identified peptides? This is more relevant than the number of identified proteins, please specify.
III) In order to make it easier to understand the details of the analysis results, please upload your data on proteomeXchange or similar platform.
IV) The volcano plots Figure 2 provide p-values only with a cutoff of 0.05. The identification of DEPs should not be based on p-values only due to the multi-parameter errors accumulated this way. Please use adjusted p-values or FDR-filtered DEPs as provided by Perseus software or several R packages.
Author Response
July 21, 2021
Prof. Dr. Samuel C. Mok
Editor-in-Chief
Cancers
Manuscript Number: cancers-1283746
Title: Mass Spectrometry-based Proteomic Discovery of Prognostic Biomarkers in Adrenal Cortical Carcinoma
Dear Editor:
We appreciate that you have given us this opportunity to revise the manuscript (cancers-1283746). We have carefully considered the reviewer’s comments and suggestions that have been helpful to improve our manuscript. We have highlighted all changes in a red font in the revised manuscript and have addressed all the reviewer’s comments point by point in the attached letter.
We hope that you find our revised manuscript satisfactory for publication in Cancers.
Sincerely,
Dohyun Han, PhD
Associate Professor, Director
Proteomics Core Facility, Biomedical Research Institute, Seoul National University Hospital
101 Daehak-ro, Jongno-gu, Seoul, 03080, Republic of Korea
Tel:+82-2-2072-1719
Fax:+82-2-2072-4406
E-mail: hdh03@snu.ac.kr
Jung Hee Kim, MD, PhD
Associate Professor
Department of Internal Medicine, Seoul National University Hospital,
Seoul National University College of Medicine
101 Dae-hak ro, Jongno gu, Seoul, 03080, Korea
Phone: +82-2-2072-4839
E-mail: jhee1@snu.ac.kr/jhkxingfu@gmail.com

Reviewer 2 Report
In the era of large genomic and proteomic datasets, it has become important to mutually validate results of individual studies using data from different publicly available sources. In this manuscript, the author performed MS based proteomics to find prognostic biomarkers of Adrenal cortical carcinoma (ACC) and validated them with the mRNA expression data from The Cancer Genome Atlas data (TCGA). This research has great value to both the proteomics and the physician community, which would inspire more proteomics research with clinical application prospect and be expected to help determine the appropriate treatment plan for ACC patients. Besides these principal concerns I would like to point out several other minor points that the authors should consider:
- Authors should provide the sources of the chemical and biological reagents used in the Materials and Methods section.
- Line 26: “lipid” should be “liquid”.
- Line 109: What is the purpose of incubating samples at 95 °C for 2 h? Will it affect the stability of the sample?
- Line 138: “N-acetylation” should be “N-terminal acetylation”.
- Line 141: The author should also set a Score filter for the identified peptides.
- The criteria for significant differences in the protein expression levels in Line 151: The p value < 0.05, fold-change > 1.5 is not the same with Line 223(P<0.05; fold-change ≥ 2.0) and Line 235(two-fold difference in expression).
- Line 194: The author had better add “benign” before “adrenal adenoma”.
- Line 210: The authors use several of the following terms to describe benign patients, and it would be better if the author can keep the context consistent: “benign tumors”, “benign adrenal adenoma”, “benign adrenal tumor”.
Author Response
July 21, 2021
Prof. Dr. Samuel C. Mok
Editor-in-Chief
Cancers
Manuscript Number: cancers-1283746
Title: Mass Spectrometry-based Proteomic Discovery of Prognostic Biomarkers in Adrenal Cortical Carcinoma
Dear Editor:
We appreciate that you have given us this opportunity to revise the manuscript (cancers-1283746). We have carefully considered the reviewer’s comments and suggestions that have been helpful to improve our manuscript. We have highlighted all changes in a red font in the revised manuscript and have addressed all the reviewer’s comments point by point in the attached letter.
We hope that you find our revised manuscript satisfactory for publication in Cancers.
Sincerely,
Dohyun Han, PhD
Associate Professor, Director
Proteomics Core Facility, Biomedical Research Institute, Seoul National University Hospital
101 Daehak-ro, Jongno-gu, Seoul, 03080, Republic of Korea
Tel:+82-2-2072-1719
Fax:+82-2-2072-4406
E-mail: hdh03@snu.ac.kr
Jung Hee Kim, MD, PhD
Associate Professor
Department of Internal Medicine, Seoul National University Hospital,
Seoul National University College of Medicine
101 Dae-hak ro, Jongno gu, Seoul, 03080, Korea
Phone: +82-2-2072-4839
E-mail: jhee1@snu.ac.kr/jhkxingfu@gmail.com
* Please see the attachment

Round 2
Reviewer 1 Report
The authors have responded to all questions satisfactorily, I endorse publication of this manuscript.
Author Response
We thank very much for your generous assessment.